# Gate-controlled suppression of light-driven proton transport through graphene electrodes

S. Huang[1,2,7], E. Griffin [1,2,7] ✉, J. Cai[1,3], B. Xin [1,2], J. Tong[1,2], Y. Fu[1,2], V. Kravets[1], F. M. Peeters[4,5] & M. Lozada-Hidalgo [1,2,6] ✉

Recent experiments demonstrated that proton transport through graphene electrodes can be accelerated by over an order of magnitude with low intensity illumination. Here we show that this photo-effect can be suppressed for a tuneable fraction of the infra-red spectrum by applying a voltage bias. Using photocurrent measurements and Raman spectroscopy, we show that such fraction can be selected by tuning the Fermi energy of electrons in graphene with a bias, a phenomenon controlled by Pauli blocking of photo-excited electrons. These findings demonstrate a dependence between graphene's electronic and proton transport properties and provide fundamental insights into molecularly thin electrode-electrolyte interfaces and their interaction with light.

The defect-free basal plane of monolayer graphene is impermeable to all atoms[1,2] and ions[3], but permeable to protons, nuclei of hydrogen atoms[4,5]. Recent experiments demonstrated that the transport takes place mostly around wrinkles and nanoscale ripples in the crystal lattice, in which strain and curvature reduce the energy barrier for proton permeation[6]. Besides graphene, which is a zero-gap semiconductor, monolayers of hexagonal boron nitride (hBN), a wide-gap insulator, were shown to be even more permeable than graphene; whereas molybdenum disulphide (MoS$_2$), a heavily doped semiconductor, was found to be impermeable to protons[4]. These results suggested that there was no correlation between the crystal's proton permeability and its electronic properties and cemented the notion that the Fermi energy of electrons in the material played no role in the proton transport. This notion became contested with experiments in which graphene's in-plane electron conductivity[7] was exploited to fabricate two-dimensional proton-permeable electrodes. In the devices, protons transfer through graphene and combine with electrons to form H$_2$ molecules;[4,5,8–10] a process that is accelerated using catalytic nanoparticles[4,5,8–10] and takes place with 100% efficiency (so-called Faradaic efficiency)[4,5,8–10]. Unexpectedly, these experiments revealed that solar-simulated illumination

can enhance proton permeation through graphene electrodes by an order of magnitude[8], a phenomenon that we called the photo-proton effect. However, despite these observations, there is still no understanding of the fundamental mechanism behind this photo-effect, nor on the dependence of proton transport through graphene on the Fermi energy of electrons. In this work, we report the unexpected suppression of the photo-proton effect with a small <1 V bias when the devices are measured under infra-red illumination. We show that this arises due to a shift in the Fermi energy of electrons in graphene under the applied bias, which prevents the absorption of low energy infra-red photons and suppresses light-driven proton transport through graphene. The results provide fundamental understanding of the interaction of photo-excited electrons and protons in molecularly thin electrode-electrolyte interfaces, which we rationalise using the Gerischer model[11,12] of electrochemical interfaces.

## Results

### Device fabrication and measurements

The devices studied in this work consisted of a graphene monolayer suspended over a micrometre-sized hole etched into silicon nitride

[1]Department of Physics and Astronomy, The University of Manchester, Manchester M13 9PL, UK. [2]National Graphene Institute, The University of Manchester, Manchester M13 9PL, UK. [3]College of Advanced Interdisciplinary Studies, National University of Defence Technology, Changsha, Hunan 410073, China. [4]Departamento de Fisica, Universidade Federal do Ceara, 60455-900 Fortaleza, Ceara, Brazil. [5]Departement Fysica, Universiteit Antwerpen, Groenenborgerlaan 171, B-2020 Antwerp, Belgium. [6]Research and Innovation Center for graphene and 2D materials (RIC2D), Khalifa University, PO Box 127788, Abu Dhabi, United Arab Emirates. [7]These authors contributed equally: S. Huang, E. Griffin. ✉e-mail: eoin.griffin@manchester.ac.uk; marcelo.lozadahidalgo@manchester.ac.uk

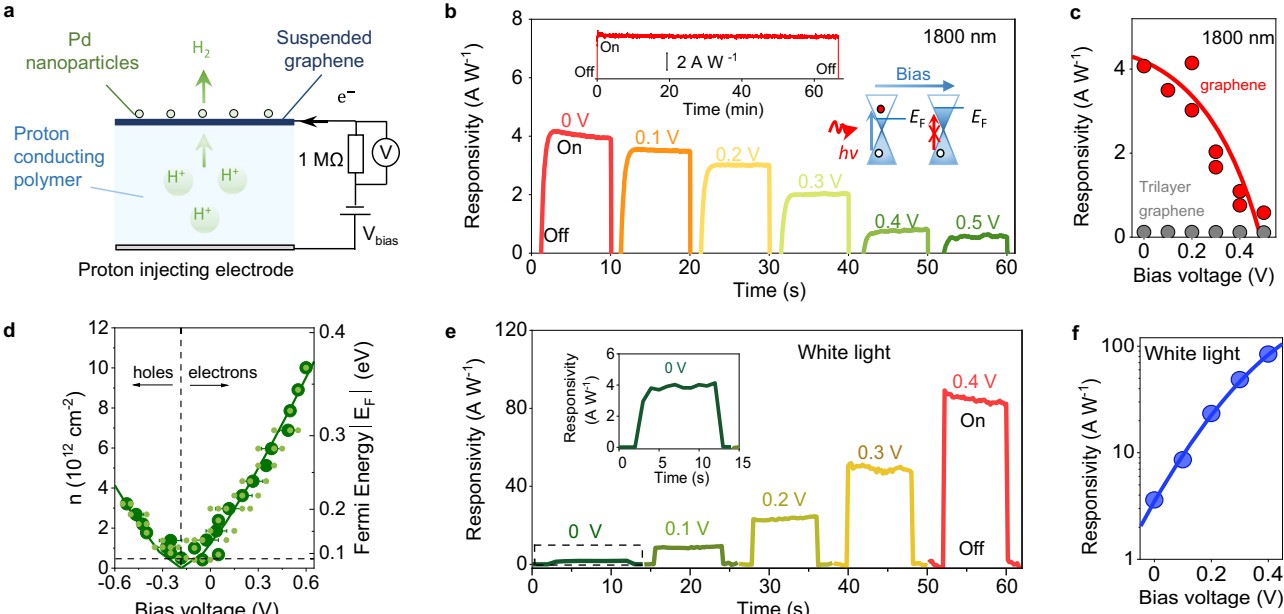

**Fig. 1 | Voltage gated suppression of the photo-proton effect. a** Schematic of experimental setup. $H_2$, hydrogen gas; $H^+$, proton; $e^-$, electrons; $V_{bias}$, applied bias. **b** Examples of responsivity vs time of devices under 1800 nm light on-off pulses shown the gate-controlled suppression of the photoresponse. In the measurement, the bias is fixed as a function of time; the light is turned on and off and the process is repeated for a different bias (colour coded). Left inset shows that the photo-response is stable for over an hour of continuous illumination of 1800 nm wave-length. $V$-bias, 0 V. Right inset, schematic illustrating the principle behind the Pauli blocking mechanism. Initially (left), $E_F$ (Fermi energy) is lower than half the photon energy (given by Planck's constant, $h$, and light frequency, $v$), allowing absorption of the photon (marked by blue arrow). When the $V$-bias is applied (right), $E_F$ rises and now blocks photon absorption (marked by crossed red arrow). **c** Red data points, responsivity vs applied bias extracted from 2 different devices under illumination of 1800 nm wavelength. Red line, guide to the eye. Grey data points, corresponding data for trilayer graphene. **d** Charge density in the graphene electrode ($n$) as a function of applied bias (left $y$-axis) and corresponding Fermi energy, $E_F$ (right $y$-axis) extracted from Raman spectroscopy measurements (Supplementary Fig. 5). Small green symbols, individual data points from the measurements. Large dark green symbols, average from small symbols. Error bars, standard deviation (SD). Solid lines, best fit of formula $E_F = \hbar v_F \sqrt{\pi n(V)}$ to data, with $n(V)$ the gate tuneable carrier density as a function of bias $V$ (Raman spectroscopy characterisation in Methods), $\hbar$, reduced Planck's constant and $v_F$, the Fermi velocity. Dotted line marks the finite doping found at the NP. **e** Examples of responsivity vs time of devices under white illumination on-off pulses. Inset, zoom in of responsivity at zero applied bias. **f** Responsivity vs applied bias extracted from (**e**). Blue line, guide to the eye.

substrates (typically 5 μm diameter), as reported previously[4,8] (Fig. 1a, Supplementary Fig. 1). One side of the suspended graphene film was decorated with Pd nanoparticles deposited by electron beam eva-poration (nominally ≈1 nm thick film) that form a discontinuous film and enhance the proton conductivity of graphene[4,8,9]. The opposite side was coated with a proton-conducting polymer (Nafion) and elec-trically connected with a proton-injecting electrode (porous carbon decorated with Pt catalyst). The whole device was placed in a gas-tight optical chamber (IR grade sapphire widow) filled with 100% $H_2$ gas at 100% relative humidity to ensure the high proton conductivity of Nafion. The chamber faced the light source and an optical chopper. The infra-red light source used was either a broadband Bentham IL1 halogen light with an electrically controlled monochromator or indi-vidual single frequency lasers. The suspended graphene film was electrically connected into the circuit shown in Fig. 1a and the photo-response from the sample was measured using a lock-in amplifier (Photocurrent measurements in Methods, Supplementary Fig. 2). In this method, the optical chopper's frequency (fixed at 33 Hz) is used as reference signal into the amplifier, which then filters all signals except those at the input frequency and thus measures only the photocurrent arising from the chopped light excitation. By removing all background signals, including the dark current in the sample, this method enables highly sensitive photocurrent measurements. The photocurrent was measured across the 1MΩ sampling resistance in the circuit and the responsivity of the devices is presented in A $W^{-1}$. We also measured devices under low intensity white light illumination using a Keithley 2636 sourcemeter as reported previously[8]. For reference, we mea-sured similar devices in which the freestanding film was trilayer

graphene, which absorbs light similarly to graphene[13–15] but is impermeable to protons[4].

## Gate-controlled suppression of the photo-proton effect

The mechanism for current flow in these devices was demonstrated previously[4,8,9]. In brief, under an applied bias, protons from Nafion transfer through the basal plane of graphene[3–5]. The protons then combine with electrons flowing into graphene from the electrical circuit[4,8] and form adsorbed hydrogen atoms on the Pd nanoparticles on the opposite side of the suspended graphene film (Pd + $H^+$ + $e^-$ → Pd*-H). The adsorbed protons eventually escape as $H_2$ gas through the discontinuous Pd film[4,8] (Fig. 1a), which occupies an area several orders of magnitude larger than the graphene electrode and effectively behaves as a drain reservoir for protons[3]. In previous work, we showed that illuminating similar samples with white light strongly increases proton transport through graphene electrodes—a phenomenon which we called the photo-proton effect[8]. Figure 1e, f show that our mea-surements are consistent with those previous observations. In a typical measurement, the sample was biased and the photoresponse was measured as a function of time under on-off white light pulses. We observed that the photoresponse increased ~100 times when the bias was increased to a few hundred millivolts, in good agreement with the measurements reported in ref. 8. that reported the same dependence of the photoresponse with bias. Unexpectedly, this behaviour was drastically different under infra-red illumination. This is shown in Fig. 1b for the case of 1800 nm wavelength illumination. For each applied voltage, we found that the photoresponse was stable for over an hour of continuous illumination (as long as we decided to measure,

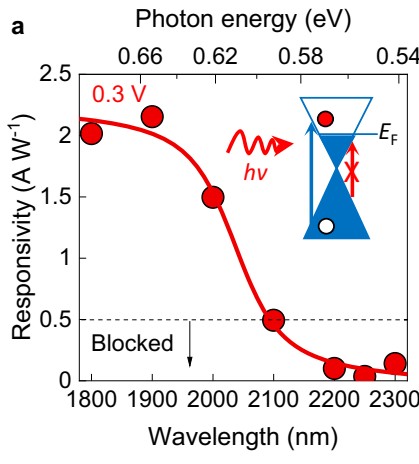

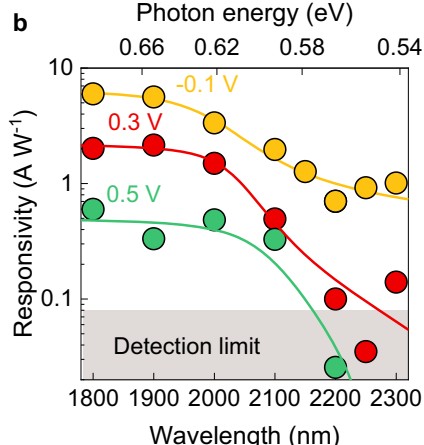

**Fig. 2 | Photon energy and voltage dependence of the suppression of the photo-proton effect. a** Example of responsivity of devices as a function of wavelength. Each data point corresponds to the average photoresponse recorded for 5 s of continuous illumination (Supplementary Fig. 7). Solid curve, guide to the eye. Dashed line, threshold for the blocking of the photoreponse. Inset, schematic illustrating the principle behind the blocking mechanism in this experiment. For a given $E_F$, which is fixed by the $V$-bias, short wavelengths with energy $E > 2E_F$ are absorbed (blue arrow), whereas longer wavelengths are blocked (red crossed arrow). **b** Examples of responsivity vs illumination wavelength for different fixed $V$-bias. The responsivity decreases with applied bias. Solid curves, guide to the eye.

inset Fig. 1b), in agreement with measurements under white light[8]. However, increasing the voltage bias did not increase the sample's photoresponse. Instead, the photocurrent was suppressed as the bias was increased to about 0.5 V. This unexpected suppression of the photoresponse was also observed with other wavelengths in the infra-red (Supplementary Fig. 3). The only difference was that the voltage at which the photoresponse was suppressed decreased with longer wavelengths, such that, for example, the photoresponse from 2000 nm light could be suppressed with ≈0.4 V (Supplementary Fig. 3). Notably, this effect is only observed in the presence of proton transport through graphene. This is evidenced from reference devices made from trilayer graphene, which absorb light similarly to graphene[13–15] but are impermeable to protons and yielded negligible photoresponse (Fig. 1c and Supplementary Fig. 4 and Photocurrent measurements in Methods).

To rationalise these observations, we note that a necessary condition for the photoresponse is the absorption of photons by graphene, which requires exciting electrons from the valence to the conduction band (Fig. 1b inset). These electrons then react with protons as described above. However, because of graphene's linear spectrum, only photons with energy ($E$) at least twice larger than the Fermi energy, $E \geq 2E_F$, can be absorbed[16]. Photons with lower energy are not absorbed because the states accessible to the photo-excited electron are occupied, a phenomenon known as Pauli blocking (Fig. 1b inset)[16,17]. On the other hand, we note that a voltage bias applied to graphene through the polymer electrolyte acts as a gate voltage[9], similar to the case of a voltage applied between graphene and a dielectric substrate[7,18]. A positive (negative) voltage dopes graphene with electrons (holes) and shifts the Fermi energy, $E_F$, with respect to the charge neutrality point (NP) according to the relation:[7] $E_F = \hbar v_F \sqrt{\pi n}$, with $n$ the charge carrier density, $v_F \approx 1 \times 10^6$ m s$^{-1}$ the Fermi velocity in graphene and $\hbar$ the reduced Planck constant[7]. Hence, applying a voltage bias raises $E_F$ and this will block the absorption.

To verify if Pauli blocking could be behind our observations, we determined $E_F$ in our graphene electrodes as a function of applied bias using Raman spectroscopy (Raman spectroscopy characterisation in Methods). Figure 1d and Supplementary Fig. 5 show that the applied $V$-bias shifts the Fermi energy of the electrodes with respect to the NP. The shifts we observe, of a couple hundred meV (corresponding to high doping of $10^{12}$–$10^{13}$ cm$^{-2}$), are sufficient to block the absorption of photons in the near infra-red. For example, the suppression of the photoresponse with 1800 nm (≈0.69 eV) light in Fig. 1a would require

$E_F \approx 0.34$ eV. The Raman data show that this $E_F$ is indeed achieved in our devices for ≈0.55 V within our experimental scatter of about ±50 mV, in agreement with our photocurrent experiments in Fig. 1b. The scatter arises from device-to-device variability in the position of the NP, which can be attributed to impurities or strain that introduce electron-hole puddles[19] and result in a finite doping of -5 × 10$^{11}$ cm$^{-2}$ at the NP (which is around −0.18 V) evident in the Raman data[19].

The unexpected voltage-gated suppression of the photo-proton effect merits additional characterisation. In the experiments described above, photo-induced proton-electron transport was suppressed by raising $E_F$ in graphene for a fixed photon energy. However, in principle, the suppression should also be achieved for a fixed $E_F$ by decreasing the photon energy until the condition $E = 2E_F$ is satisfied (inset Fig. 2a). To investigate this possibility, we measured the photoresponse of our samples using variable illumination wavelength ranging from 1800–2300 nm under fixed $V$-bias. The power density of the light of each wavelength was characterised (typically -0.1 mW cm$^{-2}$) and, from the linear power dependence of the devices' photoresponse, we extracted their photoresponsivity for each wavelength (Supplementary Fig. 6 and Photocurrent measurements in Methods). The key finding from these measurements is illustrated in Fig. 2a and Supplementary Fig.7, which show that for fixed $V$-bias the photoresponsivity decreased by an order of magnitude as the wavelength increased from 1800–2300 nm. The suppressed photoresponse corresponds to just 1% of that observed under white light for the same $V$-bias. From these observations, we consider that a device's photoresponse is blocked if the photoresponse reaches 1% of that under white light for the same $V$-bias (99% suppression).

Figure 2b shows that the photoresponse as a function of wavelength in the infra-red decreases with applied $V$-bias. This leads to shorter cut-off wavelengths for larger $V$-bias, as per the criterion defined above (99% suppression). Figure 3a shows the relation between cut-off wavelength (left $y$-axis), its corresponding photon energy $E$ (right $y$-axis) and applied $V$-bias extracted from all our measurements, at both constant bias (variable wavelength, Fig. 2a, b) and constant wavelength (variable voltage, Fig. 1). Both sets of data are consistent with each other and show that the photon energy, $E$, at which the photoresponse is blocked can be tuned by about 0.3 eV with $V$-bias. This tuning yields $E_F(V) = E(V)/2$, which can be compared with the $E_F(V)$ relation found in the Raman data (Fig. 1d). Figure 3a shows that the $E_F(V)$ Raman data from Fig. 1d is in good agreement with our photocurrent measurements within our experimental accuracy. The

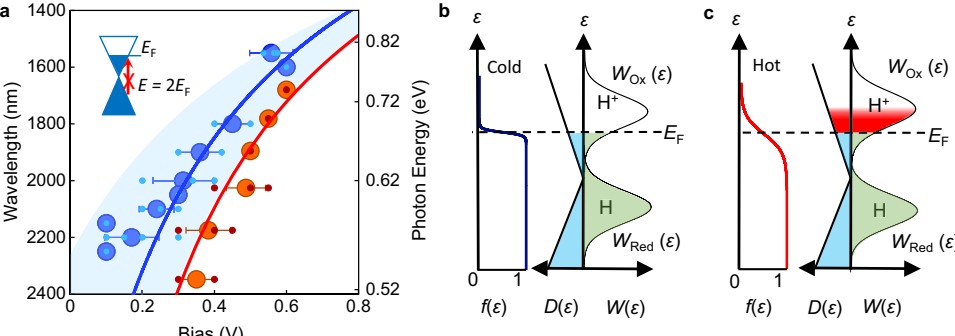

**Fig. 3 | Cut-off wavelength as a function of applied bias and Gerischer model.** **a** Cut-off wavelength as a function of applied bias for all our measurements both at constant illumination wavelength and constant applied bias (11 devices shown in total). Right $y$-axis, corresponding photon energy ($E$). Small light blue symbols, individual data points from the measurements. Large dark blue symbols, average of light blue symbols for each wavelength. Error bars, SD. Red symbols, Raman data from Fig. 1d. Both photocurrent and Raman data are fitted with $E_F = \hbar v_F \sqrt{\pi n(V)}$, using the formulae except the NP is shifted by $\approx -100$ mV in the photocurrent data (Raman spectroscopy characterisation in Methods). Shaded blue area marks the range of $E_F(V)$ fittings consistent with the data. Inset, schematic of Pauli blocking condition, $E = 2E_F$. **b** schematic of the Gerischer model for graphene electrodes under dark conditions. Graph on the left of schematic, Fermi distribution, $f(\varepsilon)$. In

dark conditions ($T_e \approx 300$ K), $f(\varepsilon)$ drops sharply at $E_F$ (marked with dashed black line). $f(\varepsilon)$ determines the filling of the density of states function in graphene, $D(\varepsilon)$. Filled states, shown in blue. On the electrolyte side, the solution density of states is represented by two Gaussian functions for the reduced (H atoms) and oxidised (proton) species. Transfer events across the interface are possible if there is an overlap in energy of filled electronic states in graphene (shown in blue) and states in solution. The overlap is shown in green. **c** corresponding schematic for the system under illumination. In this case, $T_e \sim 1000$ K, for which $f(\varepsilon)$ drops over a larger energy range for a given $E_F$. This leads to more electronic states being filled, which increases the overlap between electronic and solution states. The additional overlap is shown in red.

only difference is a relatively small shift ($\approx$100 mV) of the extracted NP in the photocurrent data, which is hardly surprising as the scatter of both Raman and photocurrent data is greatest closer to the NP. Hence, both photocurrent and Raman spectroscopy data show that the photo-proton effect is suppressed for high $E_F$, which demonstrates that proton-electron transport in the devices depends on the Fermi energy in graphene.

## Theory model of the photoresponse

To understand this connection between Fermi energy and proton-electron transport, we use the Gerischer model[11,12], which considers the role of the band structure of the electrode in the reactivity of an electrochemical interface[11] (Figs. 3b and 3c). The model describes only the rate of electron transfer. However, since proton transport in graphene electrodes is accompanied by electron transfer (the Faradaic efficiency of the process is 100% both in dark conditions and under illumination)[4,8], we use the model to gain insights into our system. The basic idea of the model is that an electron can transfer from a donor state in the electrode to an accepting state in the reactant if both states have the same energy (Gerischer model in Methods). The rate of transfer is then proportional to the integral of all the possible transfer events across all energy states. For the case of proton transport through graphene accompanied by an electron transfer (reduction process), the rate is:[11] $k \propto \nu \int_{-\infty}^{\infty} D(\varepsilon) f(\varepsilon, T_e) W_O(\varepsilon)\, d\varepsilon$. In the equation, $\varepsilon$ is the energy of the electrons; $D(\varepsilon)$ the electron density of states in graphene; $f(\varepsilon, T_e)$ the Fermi distribution; $T_e$ the temperature of the electronic system; $\nu$ a constant that determines the timescale of the reaction. $W_O(\varepsilon)$ is a Gaussian function that models the energy distribution of electron accepting species; in this case, the protons. Figure 3b, c illustrates that, while the integral is calculated over the whole energy range, the integrand is only non-zero if there is an overlap between the energy of occupied states in the electrode and accepting states in the reactant. Larger overlaps result in a larger reaction rate.

The key insight from this model is the elucidation of the role of the Fermi distribution and the electronic temperature in our devices' photoresponse. Illuminating graphene is known to create a long lived (>1 ps) hot electron distribution with high $T_e$ that can reach ~1000 K (in dark conditions, $T_e = 300$ K, Gerischer model in Methods)[20–22]. According to the Fermi distribution, electronic states with energy of up

to ~$4k_B T_e$ above $E_F$ are filled ($k_B$ the Boltzmann constant), so raising $T_e$ increases the number of filled electronic states in graphene for a given $E_F$. Since the hot electron distribution is effectively constant (>1 ps lifetime) for the duration of a proton transfer event (~$10^{-13}$–$10^{-14}$ s, refs. 23–35.), the electronic levels filled with hot electrons can interact with the solution states. This leads to a larger overlap between electronic and solution states, and hence to a faster reaction rate, $k$ (Fig. 3b, c). To validate this model, we used it to obtain order-of-magnitude estimates for the photoresponse with a typical electronic temperature in graphene under illumination of refs. 20–22 $T_e = 500$–1000 K (300 K in dark conditions). The model predicts that for such $T_e$ the reaction rate at a given $E_F$ should increase by an order of magnitude with respect to the reaction in the dark ($T_e = 300$ K), in agreement with our observations in Fig. 1d and ref. 8. (Gerischer model in Methods). The model also predicts that such photoresponse should increase exponentially with bias, in agreement with our measurements in Fig. 1e, f (Supplementary Fig. 8) and ref. 8. Finally, the model naturally incorporates Pauli blocking of the photo-proton effect. If photon absorption is blocked, $T_e$ does not rise and the reaction rate does not increase under illumination either. Hence, this model is consistent with all our observations.

## Discussion

Our experiments reveal insights into proton-electron transport across one-atom-thick interfaces, which are increasingly explored due to their unique catalytic properties[9,26,27]. We propose that absorbed photons generate a hot electron distribution in graphene that increases the overlap in energy between electronic and solutions states, leading to a larger number of possible proton-electron transfer events and thus to larger interfacial reactivity, as predicted by the Gerischer model. In the wider context of ion transport phenomena, these results show that protons interact with the electronic system in graphene under a large range of Fermi energies, unlike, for example, metal electrodes in which the Fermi energy is effectively constant for the entire operational voltage window[28]. Our results also provide insights into molecularly thin electrode-electrolyte interfaces and their interaction with light[29,30], revealing the prominent role of the electron Fermi distribution and electronic temperature. In terms of applications, graphene electrodes are a rare example of a system in which an external

stimulus, in this case a voltage, can suppress light driven proton transport for a controllable fraction of the light spectrum. These results could enable the design of light-powered devices in emerging technologies, such as proton-based neuromorphic hardware[31–35], by providing an additional gating mechanism that could enable switching individual elements of these systems or more complex logic operations.

## Methods

### Device fabrication and measurement setup

Monocrystalline graphene was obtained from graphite crystals by mechanical exfoliation and suspended over a hole etched into silicon nitride substrates (typically 5 μm diameter)[4]. The substrate fabrication follows the recipe in previous reports[4]. In brief, silicon nitride wafers (500 nm $SiN_x$ on B doped Si, purchased from Inseto Ltd.) were first patterned by photolithography and reactive ion etching (RIE) is then used to remove a $0.8 \times 0.8$ mm$^2$ section from one of the SiNx layers. The wafer is placed in the 30 wt.% KOH solution to etch out Si, leaving a free-standing SiNx window ($100 \times 100$ μm$^2$). A circular hole (typically 5 μm diameter) was patterned in the centre of the $SiN_x$ window by photolithography and etched by RIE. The suspended graphene membrane was electrically connected with an Au electrode fabricated using photolithography and electron-beam evaporation (Supplementary Fig. 1). One side of the suspended membrane was decorated with Pd nanoparticles deposited via electron-beam evaporation (nominally 1 nm thick). This process creates a discontinuous Pd film that allows the generated $H_2$ to escape. The opposite side of the membrane was coated with Nafion polymer, as previously reported[4]. In the devices, protons transfer from Nafion through graphene, then combine with electrons and adsorb on the Pd ($H^+ + e^- \rightarrow$ Pd*-H), a process accelerated by nanoscale corrugations[6] in graphene and the catalytic activity of Pd. The Nafion was electrically contacted with a porous carbon electrode loaded with Pt catalyst. For measurements, the whole device is placed inside an air-tight metal chamber with a thin IR grade sapphire window. The chamber is purged with humid $H_2$ gas to ensure the high proton conductivity of the devices.

### Photocurrent measurements

The chamber containing the sample was mounted on a stage with micro-manipulators. The window in the chamber faced a black tube with a small aperture to block stray light and the opposite end of this tube faced a light chopper (Thorlabs MC2000B). The light sources used were either a Bentham IL1 halogen broadband source or individual frequency lasers (Thorlabs NIR laser series). For measurements with the Bentham system, the light was directed through an electrically controlled single monochromator (Bentham TMc300), which allowed selecting a particular wavelength from 1800 nm to 2500 nm with a resolution of 0.2 nm. To remove higher-order wavelengths from the diffracted light, we used a 1550 nm long pass filter. For measurements with both individual lasers and the Bentham system, the sample was connected into the circuit shown in Supplementary Fig. 2, which contained a 1 MΩ resistor in series with the device. This resistance is much smaller than the resistance of the device (-10$^8$ to 10$^9$ Ω for the V-bias we used in this work), which allows measuring the photocurrent from the sample by measuring the voltage drop across this resistor. The sample was biased with a Keithley source metre (2614B) and the photocurrent was measured with a lock-in amplifier (Stanford Research Systems SR865A) using the frequency of the chopper as reference signal[36,37].

The use of the lock-in amplifier is an important difference of the measurement setup in this work compared to refs. 8. This instrument uses the light chopper's frequency as input and filters out all signals that do not have this frequency, including the dark current. This allows us to measure the photocurrent directly. If we were to measure without the lock-in amplifier, we would observe that the dark current rises with bias, as in refs. 8,10. However, the low intensity light of <1 mW cm$^{-2}$ used here would yield only a small rise in the total current, that would be difficult to characterise accurately. The lock in amplifier allows us to clearly characterise even this small photo-response.

We performed two photocurrent measurement experiments with infra-red light. In the first one, we measured the photocurrent of the samples with variable voltage bias (constant illumination wavelength). In the second experiment, we measured the photocurrent at constant voltage bias (variable illumination wavelength). For this experiment, the wavelength is fixed (within 0.2 nm) for various values between 1800–2300 nm in the Bentham system. To account for variations in the intensity of the light filtered at different wavelengths, we characterised the power density for each wavelength (Thorlabs S401C−Thermal Power Sensor Head, Surface Absorber, 0.19-20 μm) and for different light intensities (the intensity is controlled by the diaphragm in the system). Supplementary Fig.6 shows that the photocurrent depended linearly on power density, from which we extracted the spectral dependence of the devices' responsivity. For measurements with white light (Oriel Sol3A solar simulator), devices were characterised with a Keithley sourcemeter, as previously reported[8].

### Raman spectroscopy characterisation

For Raman measurements, the samples were placed in the gas-tight optical cell. The Raman spectra was measured using a WITec alpha300 R−Raman microscope with a 514 nm laser. In the experiments, the position of the $G$-band of graphene is traced as a function of applied bias (see Supplementary Fig. 5). The shift in the $G$ band ($\Delta\omega_G$) reveals the Fermi energy of electrons in graphene and the charge carrier concentration ($n$) via the following relations[38]. For electrons, $E_F$ [meV] = 21 $\Delta\omega_G$ + 75[cm$^{-1}$], for holes, $E_F$ [meV] = −18 $\Delta\omega_G$ − 83[cm$^{-1}$]. Both fittings are consistent with our data within the experimental scatter. For both type of carriers, their density is given by $n$[cm$^{-2}$] = $(E_F/11.65)^2$ x 10$^{10}$. We fit the data with the formula $E_F = \hbar v_F \sqrt{\pi n(V)}$, which $n(V)$ is gate tuneable carrier density ($n$) as a function of bias ($V$). The relationship between bias ($V$) and charge carrier density ($n$)[18] is given by $|V - NP| = \frac{\hbar v_F \sqrt{\pi n}}{e} + \frac{ne}{C}$ with the fitting parameters $C \approx 3.6$ μF cm$^{-2}$, the gate capacitance and NP = -0.18 V, the neutrality point.

### Gerischer model

The Gerischer model is evaluated as follows. On the electrode, the number of electronic states is given by the density of states function for graphene[39], $D(\varepsilon)$. On the other hand, states in solution are modelled by two density functions, $W_O(\varepsilon)$ and $W_R(\varepsilon)$, corresponding to the two different chemical species on the surface of the electrode: the proton ($W_O$) and the adsorbed hydrogen atom ($W_R$) in our case. Each of these functions is a Gaussian with mean energy equal to $\varepsilon_0 + \lambda$ for the reduction states, $\varepsilon_0 - \lambda$ for the oxidation states, and standard deviation $(2\lambda k_B T)^{1/2}$. In here, $\varepsilon_0$ is the standard potential for the reaction, $k_B T$ the thermal energy in the solution and $\lambda$ is the so-called reorganizational energy, which is the energy required for the reactants to undergo the physical transformation involved in the reaction (e.g. change of coordinates). To evaluate the model, we estimate $\varepsilon_0$ and $\lambda$. For $\varepsilon_0$ we take $\varepsilon_0 = 0$ V vs SHE (standard hydrogen electrode) for the hydrogen evolution reaction. This potential corresponds to $\varepsilon_0^{abs} \approx 4.44$ V in the so-called absolute scale[11], in which the energy of electrons is referred against their energy in vacuum.

Since in our experiments we determine the position of the neutrality point (NP) experimentally, we refer $\varepsilon_0$ vs the NP ($\varepsilon_0^{NP}$), rather than vs vacuum (Supplementary Fig. 8). To that end, we note that the energy necessary to remove an electron from the neutrality point in graphene into the vacuum is[40] $F^G \approx 4.5$ eV; and hence we estimate that $\varepsilon_0^{NP} = F^G - \varepsilon_0^{abs}$ should be around 50 meV. On the other hand, estimating $\lambda$ is difficult even for well-established reactions[11] and an accurate

estimate of this parameter is beyond the scope of this work. However, we note that $\lambda$ is expected to be larger than the activation energy for the reaction[11], which for proton transport through Pd-decorated graphene is[4,8] $\approx 0.4$ eV, both in dark conditions and under illumination (illumination does not change this energy[8]). Hence, to illustrate the model, we use $\lambda = 0.5$ eV, as a representative value for this parameter. We also evaluated the model for several parameters and found that the qualitative insights are insensitive to this number, as long as $\lambda$ is larger than the activation energy for the process.

Supplementary Fig. 8 shows the rate of the reduction reaction (proton-electron transfer) as a function of Fermi energy evaluated from the integral $k \propto \int_{-\infty}^{\infty} D(\varepsilon) f(\varepsilon, T_e) W_o(\varepsilon, \lambda)\, dE$ using the parameters described above. The rate is normalised against the reaction rate calculated for $E_F = 0$ in dark conditions. We find that the rate increases exponentially with $E_F$, both in dark conditions and under illumination, in agreement with our experimental observations. The model also predicts that for an electronic temperature of $T_e = 1000$ K, the reaction rate should increase by over an order of magnitude, in agreement with our observations in Fig. 1e and ref. 8. We note that the value of $T_e$ arises from a chain of complex processes involving photoexcitation, thermalisation and cooling[20], which can be expected to be different in our devices due to the interaction of protons with the electron cloud and the presence of noble metal nanoparticles (e.g. Pd)[41,42]. Given these two uncertainties, we ran the model using $T_e = 1000$ K, a value typically reported in the literature for graphene devices[20]. However, we also ran the model using lower $T_e = 500$ K. This lower temperature does not change the qualitative findings of the model, requiring relatively minor adjustments to the parameters to yield similar results ($\lambda = 1$ eV, $\varepsilon_0 = 100$ meV, $T_e = 500$ K).

## Data availability
Relevant data supporting the key findings of this study are available within the article and the Supplementary Information file. All raw data generated during the current study are available from the corresponding authors upon request.

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

## Acknowledgements

This work was supported by The Royal Society (URF\R1\201515, M.L.-H.) and Engineering and Physical Sciences Research Council (EP/X017745, M.L.-H.). S.H. would like to thank the Swiss National Science Foundation Postdoc Mobility Fellowships for funding.

## Author contributions

M.L-H. designed and directed the project with S.H and E.G.. S.H. and E.G. fabricated devices and performed measurements with help from B.X. and J.T. Y.F., V.K. and J.C. provided technical support. F.M.P. provided theory support. S.H., E.G and M.L.-H. Interpreted data and wrote the manuscript.

## Competing interests

The authors declare no competing interests.
