## [Peer Review File · Nature Communications]

Gate-controlled suppression of light-driven proton transport through graphene electrodesREVIEWER COMMENTS

Reviewer #1 (Remarks to the Author):

Huang and colleagues show proton transfer measurements at graphene electrodes and find a clear effect of changing graphene's Fermi energy and photon energy, which can be explained by the well-known Pauli blocking effect. These are interesting results that seem to confirm the role of light-induced e-h pair creation in graphene towards proton transfer reactions, which would be a useful result. Indeed, whereas this Pauli blocking effect is not as surprising as the authors would like the reader to believe (mentioning this several times), it could provide clear signatures that the electronic properties of graphene play an important role in proton transfer, and this would merit publication.

The results, however, do rely on one important assumption, which is that there is no light-induced response ending up in their photocurrent signal that does not correspond to proton transfer. I would recommend the authors to perform such a check measurement, for example with a modified Nafion membrane.

Another comment is that the authors assume a hot electron temperature of ~ 1000 K, but this should be depend on fluence (non-linearly). How did the authors estimate this temperature? And why did they use linear fits to their power dependence?

Reviewer #2 (Remarks to the Author):

This work reveals an interesting photoelectric property of graphene that the light-driven proton transport through graphene electrodes is suppressed under infrared illumination with a small voltage bias. This observation is attributed to Pauli blocking and can be elucidated by the Gerischer model. The study is thorough, and the data presented are consistent with the claims. The role of the shift in the Fermi energy of electrons in graphene under the applied bias on the device's photoresponse is convincing and supported by a theoretical model. Thus, I recommend it for publication after the author address the below concerns. revisions.

1) Is the voltage gate-controlled suppression of the proton-electron transport reversible?

2) Are there any other factors besides the Fermi distribution and the electronic temperature that can influence this photoresponse phenomenon?

3) What are the potential applications of this unique voltage gate-controlled photoresponse of graphene? The photoresponse performance in terms of response speed, photostability, broadband spectrum responsivity needs to consider for meeting the requirement of practicability.

Reviewer #3 (Remarks to the Author):

This manuscript describes an unexpected phenomenon that the light-driven proton transport through graphene electrodes will be suppressed for a tunable fraction of the infrared spectrum by applying a voltage bias. The result significant departure from the previous research that low-intensity illumination will accelerate proton transport through graphene electrodes over an order of magnitude.

According to the photocurrent measurements and Raman spectroscopy, the bias on the graphene electrodes will influence the Fermi energy of electrons and thus prevents the absorption of infrared photons, known as Pauli blocking. Meanwhile, the Gerischer model is used to elucidate the role of the Fermi distribution and the electronic temperature in the devices' photo response, which provide new insights into molecularly thin electrode-electrolyte interfaces and their interaction with light.

The research results provide a fundamentally new understanding of the interaction of photo-excited electrons and protons in molecularly-thin electrode-electrolyte interfaces and facilitate to improve the understanding of the catalytic mechanism of molecular-thin materials.

This manuscript is suggested to be published in Nat. Commun. after minor revision.

Relevant comments are given as follows:

1. In Fig. 1a, according to the schematic of the experimental setup, Pd nanoparticles were chosen to decorate the graphene film, some background information on why Pd nanoparticles were chosen is suggested to be given. Meanwhile, a detailed procession that how the proton transport the graphene is suggested to be given.

2. In Fig. 1b and Fig. 1c, under the 1800 nm light on-off pulses, with the increase of the bias from 0 to 0.5 V, the responsivity of the device gradually tapers off. However, according to the Wien effect and the data

in previous research, (Nature Nanotech 2018, 13, 300), in dark conditions, the Current density gradually rises with the improvement of bias, please explain the reason for the opposite results in the two studies.

3. The previous experiments showed that proton transport through graphene electrodes is strongly enhanced under illumination via a hot-electron-mediated mechanism, the so-called photoproton effect mechanism. Compare with white light, Infrared light has a lower photon energy. The absorption of infrared light by graphene is more tend to cause the transition of electrons from the conducting band to the valence band. So it is recommended to explain where the difference between the change in wavelength and the absorption of graphene is.

Reply to comments from Reviewer #1

Huang and colleagues show proton transfer measurements at graphene electrodes and find a clear effect of changing graphene's Fermi energy and photon energy, which can be explained by the well-known Pauli blocking effect. These are interesting results that seem to confirm the role of light-induced e-h pair creation in graphene towards proton transfer reactions, which would be a useful result. Indeed, whereas this Pauli blocking effect is not as surprising as the authors would like the reader to believe (mentioning this several times), it could provide clear signatures that the electronic properties of graphene play an important role in proton transfer, and this would merit publication. We thank the Reviewer for this positive assessment of our work.

The results, however, do rely on one important assumption, which is that there is no light-induced response ending up in their photocurrent signal that does not correspond to proton transfer. I would recommend the authors to perform such a check measurement, for example with a modified Nafion membrane.

We completely agree with the Reviewer. Following their advice, we measured devices fabricated in the same way, except monolayer graphene was substituted for trilayer graphene. Trilayer graphene absorbs light similarly to graphene^{1,2,3} and, crucially, is impermeable to protons (ref. 4 in manuscript). Hence, trilayer devices allow us to probe all the current that does not correspond to proton transport through graphene. We found that trilayer devices display current 2 orders of magnitude smaller than monolayer graphene in dark conditions. Next, we measured the devices under infrared light of a fixed wavelength with the lock-in amplifier. This technique, which removes the dark current, allows us to pick up even tiny photocurrent signals. Fig. R1 shows that it is possible to discern a very small photocurrent in this experiment. However, this photocurrent is over an order of magnitude lower than that from monolayer devices and is within the current level at which we would normally consider the device as blocked. Crucially, this tiny signal displays no dependence on applied bias, unlike the photo-response in monolayer devices, which displays a sharp dependence on bias. Hence, these findings confirm that the phenomena we observe in monolayer devices is due to proton transport through graphene and that parasitic process contribute negligibly to our observations. We are sincerely grateful with the Reviewer for pointing this to us, we believe the manuscript has been meaningfully improved with this new experiment. This experiment is now included in the results section in Fig. 1c and in Fig. S4 in the revised manuscript.

Fig. R1 | Trilayer graphene devices show voltage-independent responsivity that is an order of magnitude lower than for monolayer devices. a, Responsivity of monolayer graphene device measured under illumination of 1800 nm wavelength. **b**, Responsivity from a trilayer graphene device measured under illumination of 1800 nm wavelength. The data show that trilayer graphene devices have negligible responsivity that does not depend on applied bias.

1. C. Hung Liu, *et al.* Observation of an electrically tunable band gap in trilayer graphene. *Nat. Physics* 7, 944-947 (2011).
2. S. Sun, *et al.*, Ultrafast hot-carrier-dominated photocurrent in graphene. *Nat. Nano.* 7, 114-118 (2012).
3. M. Kim, *et al.*, Photocurrent generation at ABA/ABC lateral junction in tri-layer graphene photodetector. *Carbon* 86, 454-458 (2016).

Another comment is that the authors assume a hot electron temperature of ~ 1000 K, but this should be depend on fluence (non-linearly). How did the authors estimate this temperature? And why did they use linear fits to their power dependence?

Hot carriers in graphene are equilibrated via a chain of complex processes involving photoexcitation, thermalisation and cooling¹. This chain of processes can be expected to be different in our devices for two reasons. First, protons pierce graphene's electronic cloud and then recombine with the photoexcited electrons to form hydrogen atoms, which could potentially affect the electron cooling process. Unfortunately, there are no experiments (e.g. ultrafast pump-probe spectroscopy) or theoretical models that can shed light into the electron thermalisation and cooling process in graphene in the presence of proton transport. Second, the electronic temperature in graphene can also be affected by the presence of noble metal nanoparticles (e.g. Pd), which can inject hot electrons and also increase the amount of absorbed light in graphene^{3,4}. Given these two unknowns, we limited ourselves to running the Gerischer model using plausible T_e values obtained from the literature¹. Following the Reviewer comment, we have also run the model using $T_e = 500$ K. We find that this lower temperature does not change the qualitative findings of the model, requiring relatively minor adjustments to the parameters to yield similar results ($\lambda = 1$ eV, $\epsilon_0 = 100$ meV, $T_e = 500$ K).

Regarding the use of a linear fit to model the power dependence in Supplementary Fig. 6, we note that our previous work² showed that the photo-response depends non-linearly on power density. However, the same data showed that for low illumination power density of < 5 mW cm⁻², the dependence becomes linear. The data in the present work was collected with power density of < 1 mW cm⁻² and is also seen to depend linearly on power, in agreement with the previous work². Considering the new insights gained in this work, this linear dependence can now be traced back to the proton-electron combination mechanism in the Gerischer model. For low illumination power density, T_e and hence the overlap between electronic and solution states can be expected to be small. In this regime, the Gerischer model is then in the linear regime, which translates into a linear power dependence for low illumination power density. Following the Reviewer comment, this discussion is now included in the Methods section in the revised manuscript, page 11, paragraph 1.

1. Massicotte, M., Soavi, G., Principi, A. & Tielrooij, K.-J. Hot carriers in graphene – fundamentals and applications. *Nanoscale* 13, 8376–8411 (2021).
2. Lozada-Hidalgo, M. et al. Giant photoeffect in proton transport through graphene membranes. *Nat. Nanotechnol.* 13, 300–303 (2018).
3. M. L. Brongersma et al. Plasmon-induced hot carrier science and technology. *Nat. Nano.* 10, 112770 (2022)
4. B. Du, et al. Plasmonic hot electron tunnelling photodetection in vertical Au-graphene hybrid nanostructures. *Laser & Photonics Rev.* 11, 1600148 (2017).

Reply to comments from Reviewer #2

This work reveals an interesting photoelectric property of graphene that the light-driven proton transport through graphene electrodes is suppressed under infrared illumination with a small voltage bias. This observation is attributed to Pauli blocking and can be elucidated by the Gerischer model.

The study is thorough, and the data presented are consistent with the claims. The role of the shift in the Fermi energy of electrons in graphene under the applied bias on the device's photoresponse is convincing and supported by a theoretical model. Thus, I recommend it for publication after the author address the below concerns. revisions.

We are very grateful to the Reviewer for this encouraging review of our work. We explain the revisions made to the manuscript to address the Reviewer comments.

1) Is the voltage gate-controlled suppression of the proton-electron transport reversible?

The gate-controlled suppression is fully reversible. The suppression arises from a shift in the Fermi level in the devices under an electric field. Switching the field off brings the Fermi level to its original position and thus reverses the suppression. This can be appreciated from our experiments in Fig. 2. In those experiments, the V -bias is fixed, and the illumination wavelength is increased until the suppression of the photo-effect takes place. We then switch off the light, change the V -bias and repeat the experiment until the suppression takes place again, this time for a different wavelength. This demonstrates that the suppression is not only reversible, but also tuneable. Following the Reviewer's comment, we have clarified this important point in page 3, Fig. 1 caption in the revised manuscript.

2) Are there any other factors besides the Fermi distribution and the electronic temperature that can influence this photoresponse phenomenon?

Besides the Fermi distribution and electronic temperature, the devices' photo-response depends sensitively on their proton conductivity. As discussed with Reviewer #1, this is evident from new control experiments using devices fabricated in the same way, except monolayer graphene was substituted for trilayer graphene. Trilayer graphene absorbs light similarly to graphene but, crucially, is impermeable to protons (ref. 4 in manuscript). Hence, these devices allow us to probe all the current that does not correspond to proton transport through graphene. We find that trilayer devices show negligible photoresponse, confirming that proton transport through graphene is essential to our observations (see question 1 from Reviewer #1). Following the Reviewer's comment, we have included this point in page 3, end of paragraph 1 in the revised manuscript.

3) What are the potential applications of this unique voltage gate-controlled photoresponse of graphene? The photoresponse performance in terms of response speed, photostability, broadband spectrum responsivity needs to consider for meeting the requirement of practicability.

While our work focuses on the fundamental understanding of light-driven proton permeation through graphene electrodes, we believe there is potential for applications. We are inspired by various recent studies, such as¹⁻⁵, that show that ion-conducting devices have potential in neuromorphic and in-memory computing applications. We imagine that our devices could be a component in novel bioinspired optical sensors based on ions and protons. Further studies are necessary to prove the suitability of the devices in such applications. Nevertheless, both our previous work (ref. 8 in the manuscript) and this work, show that these devices show promise. For example, in ref. 8 we showed that the devices are stable for hours of continuous illumination (also demonstrated in the present work), can endure $>10^6$ on-off cycles without degradation and respond to illumination in the tens of micro-second timescales. In this work, we now show that the devices can operate in the wavelengths used for optical fibre communication applications (e.g. 1550 nm) and we demonstrate that their response to these wavelengths can be tuned with a gate voltage. These devices could therefore enable

coupling ion-based devices and optoelectronic signals. Following the Reviewer comment we discuss this point in the 'Outlook' section of the revised manuscript.

1. Onen, M., Li, J., Yildiz, B. & Del Alamo, J. A. Dynamics of PSG-Based Nanosecond Protonic Programmable Resistors for Analog Deep Learning. *Science* 377, 539–543 (2022).
2. Shao, Z. et al. All-solid-state proton-based tandem structures for fast-switching electrochromic devices. *Nat. Electron.* 5, 45–52 (2022).
3. He, X. et al. Proton-mediated reversible switching of metastable ferroelectric phases with low operation voltages. *Sci. Adv.* 9, eadg4561 (2023).
4. Zhu, L. Q., Wan, C. J., Guo, L. Q., Shi, Y. & Wan, Q. Artificial synapse network on inorganic proton conductor for neuromorphic systems. *Nat. Commun.* 5, (2014).
5. Robin, P. et al. Long-term memory and synapse-like dynamics in two-dimensional nanofluidic channels. *Science* 379, 161–167 (2023).

Reply to comments from Reviewer #3

This manuscript describes an unexpected phenomenon that the light-driven proton transport through graphene electrodes will be suppressed for a tunable fraction of the infrared spectrum by applying a voltage bias. The result significant departure from the previous research that low-intensity illumination will accelerate proton transport through graphene electrodes over an order of magnitude.

According to the photocurrent measurements and Raman spectroscopy, the bias on the graphene electrodes will influence the Fermi energy of electrons and thus prevents the absorption of infrared photons, known as Pauli blocking. Meanwhile, the Gerischer model is used to elucidate the role of the Fermi distribution and the electronic temperature in the devices' photo response, which provide new insights into molecularly thin electrode-electrolyte interfaces and their interaction with light. The research results provide a fundamentally new understanding of the interaction of photo-excited electrons and protons in molecularly-thin electrode-electrolyte interfaces and facilitate to improve the understanding of the catalytic mechanism of molecular-thin materials.

This manuscript is suggested to be published in *Nat. Commun.* after minor revision.

We thank the Reviewer for this positive assessment of our work.

Relevant comments are given as follows:

1. In Fig. 1a, according to the schematic of the experimental setup, Pd nanoparticles were chosen to decorate the graphene film, some background information on why Pd nanoparticles were chosen is suggested to be given. Meanwhile, a detailed procession that how the proton transport the graphene is suggested to be given.

The role of the Pd nanoparticles is weaved throughout various stages of the transport process. This can be appreciated from a more detailed description of the proton transport mechanism in the devices, as suggested by the Reviewer.

Under an applied bias, protons from Nafion transfer through the basal plane of graphene, a process that is accelerated by Pd nanoparticles. This acceleration with Pd was demonstrated in ref. 4 and the microscopic mechanism revealed in recent work (ref. 6 in the manuscript), which demonstrated that proton transport is facilitated by nanoscale corrugations, such as those induced by Pd nanoparticles, that strain the lattice and lower the barrier for proton transport. After the protons transfer through graphene, they combine with electrons, which flow into graphene from the electrical circuit and form adsorbed hydrogen atoms on the Pd nanoparticles on the opposite side of the suspended graphene film ($\text{Pd} + \text{H}^+ + \text{e}^- \rightarrow \text{Pd}^*\text{-H}$). These adsorbed H atoms eventually escape as H_2 gas through the

discontinuous Pd film, which is an excellent catalyst for this reaction. Following the Reviewer comment, the discussion on the transfer mechanism is expanded in the Methods section page 9, paragraph 1.

2. In Fig. 1b and Fig. 1c, under the 1800 nm light on-off pulses, with the increase of the bias from 0 to 0.5 V, the responsivity of the device gradually tapers off. However, according to the Wien effect and the data in previous research, (Nature Nanotech 2018, 13, 300), in dark conditions, the Current density gradually rises with the improvement of bias, please explain the reason for the opposite results in the two studies.

The difference arises because the measurement technique used in this work (lock-in amplifier) is different from the previous works. In a nutshell, the technique measures the photocurrent directly and removes the dark current. We expand on this below.

In this work the photo-response is much lower than in our previous works because of the low illumination power density used ($<1 \text{ mW cm}^{-2}$) and because of Pauli blocking. Hence, to obtain a clear response, we measured the devices using a lock-in amplifier. This instrument uses the light chopper's frequency as input and then filters out all signals that do not have this frequency, including the dark current. In this way we only measure the photocurrent. If we were to measure without the lock-in amplifier, we would indeed see that the dark current rises with bias, just as in refs. 8&9. However, shining low intensity light of $<1 \text{ mW cm}^{-2}$ would yield only a small increment in the total current, which would be prone to being smeared by noise. The lock in amplifier, which is the gold standard technique for photocurrent measurements, allows us to clearly characterise even this small photo-response. Following the Reviewer comment, this point is stressed in the caption of Fig. 1 and page in 9 paragraph 3 in the Methods in the revised manuscript.

3. The previous experiments showed that proton transport through graphene electrodes is strongly enhanced under illumination via a hot-electron-mediated mechanism, the so-called photoproton effect mechanism. Compare with white light, Infrared light has a lower photon energy. The absorption of infrared light by graphene is more tend to cause the transition of electrons from the conducting band to the valence band. So it is recommended to explain where the difference between the change in wavelength and the absorption of graphene is.

In general, the main light absorption mechanisms in graphene involve inter-band absorption and intra-band absorption¹. In inter-band absorption, a photon promotes a valance-band electron to the conduction band. This mechanism dominates whenever the photon energy is >2 times the Fermi energy of electrons in graphene. Crucially, this mechanism is sensitive to the Fermi level, so inter-band absorption is demonstrated by probing the photo-response as a function of the Fermi energy. In our experiments, we demonstrated this dependence on the Fermi level, which shows that inter-band absorption is behind our observations. On the other hand, intra-band transitions are visible if the photon energy is less than twice the Fermi energy. In this case, photo-excited carriers in either the valance (hole doping) or conduction bands (electron doping) can undergo transitions within that band. However, the photon absorption rate in this case is much lower than for inter-band absorption and we did not observe a photo response in this case.

1. Mak, K. F., Ju, L., Wang, F. & Heinz, T. F. Optical spectroscopy of graphene: From the far infrared to the ultraviolet. *Solid State Commun.* **152**, 1341–1349 (2012).

REVIEWERS' COMMENTS

Reviewer #1 (Remarks to the Author):

I appreciate the very thorough and convincing response and revision, and recommend publication of this interesting work.

Reviewer #2 (Remarks to the Author):

The authors have addressed the questions and concerns and now the manuscript can be accepted.

Reviewer #3 (Remarks to the Author):

The manuscript can be accepted as it is.